# Towards Robust Out-of-Distribution Generalization for Deep Neural Networks with Tailored Data Regularization

## Abstract

Out-of-Distribution (OOD) generalization remains both a fundamental challenge and an often-overlooked aspect of modern machine learning—especially in the context of Deep Neural Networks (DNNs), which are highly expressive yet prone to overfitting under distributional stress. Classical learning theory highlights the role of regularization in managing the bias-variance trade-off—particularly important for complex models with higher VC dimension. In this work, we explore **stochastic data regularization techniques**—such as random transformations and noise injection—applied not only as isolated strategies but also organized through a Scheduling Policy framework using a Curriculum Learning-based approach. By progressively increasing input difficulty during training, the scheduling aligns model capacity with task complexity, promoting more **robust generalization**. We also propose a novel statistical procedure to assess the consistency of performance estimates across cross-validation folds, mitigating miscoverage effects in confidence interval estimation. Altogether, our findings highlight the importance of a **tailored data regularization, where the selection, combination, and scheduling** of perturbations become key to achieving OOD robustness in DNNs.

## 1 Introduction

Robust generalization is the ability of models to maintain reliable performance under distribution shifts—when test data deviate from the training distribution. It remains a significant challenge for Deep Neural Networks (DNNs), which are highly susceptible to overfitting under distribution shifts (Li et al., 2022; Hendrycks et al., 2021). Within the broader landscape of robust training strategies, regularization techniques are commonly used to counteract this issue—but they often fall short or even cause over-regularization, degrading model performance (Lin et al., 2024). These limitations highlight the need for carefully tailored regularizers (Choi & Kim, 2024; Srivastava et al., 2014) whose effectiveness depends heavily on both the task (e.g., classification) and the data domain (e.g., vision, language). This motivates the need for domain-aware and task-sensitive regularization approaches.

In this work, we adopt the OOD definitions of Farquhar & Gal (2022), focusing on the **transformed-OOD** setting, where label-preserving corruptions are applied to in-distribution inputs. To assess the impact of such shifts, we compute statistical distances—such as KL divergence—between clean and corrupted latent representations, capturing the extent of deviation and the effect of regularization.

Building on this perspective, we explore whether stochastic data regularization—via random transformations (Cubuk et al., 2020; Hendrycks et al., 2019) and input noise (Bishop, 1995; Yuan et al., 2025; Filho et al., 2023)—can act as an implicit regularizer when applied dynamically. We show that organizing these perturbations as a curriculum (Bengio et al., 2009; Lu & Lam, 2023), gradually increasing their intensity, is a promising yet underexplored strategy in computer vision (Choi & Kim, 2024)—especially effective for compact models aiming at robust generalization.

A modular framework that systematizes data regularization is presented through three components: a *Selection Policy* (e.g., choosing between noise types or augmentation pipelines), a *Combination Policy* (e.g., composing augmentations and noise), and a *Scheduling Policy* (Curriculum Learning-based approach).

This paper explores a core question in robust model design: **How can data regularization be dynamically adapted to a model's capacity to improve robustness while mitigating overfitting and underfitting, thereby enhancing out-of-distribution performance?** We hypothesize that stochastic data regularization—whether applied uniformly or progressively—can drive consistent gains in robustness across domains. When organized as a curriculum, aligning perturbation strength with model maturity, such strategies can enhance learning stability. Moreover, even unstructured randomness in augmentations and noise appears effective in reducing overfitting and promoting generalization, particularly when carefully tuned to avoid early over-perturbation.

To evaluate our models, we assess both average performance and the reliability of performance estimates. We introduce a miscoverage-based analysis across cross-validation folds, inspired by Bates et al. (2023), to quantify how well confidence intervals reflect true variability. Our findings indicate that stronger data regularization reduces miscoverage—particularly in shallow architectures—by addressing the bias–variance trade-off and promoting more stable out-of-distribution generalization.

## 2 RELATED WORKS

**Out-of-Distribution Categorization**  The term out-of-distribution is often used ambiguously in the literature, leading to inconsistent interpretations and methodological practices. To clarify this, recent work (Farquhar & Gal, 2022) categorizes the different distributions into four types: transformed, related, complementary, and synthetic.

**Diversity in Data Regularization for Robust Learning**  Recent work has shown that data regularization plays a key role in improving both robustness and generalization (Li & Spratling, 2023). However, simple transformations are often insufficient under distribution shifts. Increasing the diversity of augmentations promotes better model performance. Diverse transformations help the model to generalize to unseen data and improve stability under adversarial conditions. This highlights the importance of carefully designed augmentation pipelines for robust learning.

**Miscoverage in Cross-Validation-Based Estimates**  Standard $K$-fold cross-validation (CV) often underestimates variance, resulting in confidence intervals with lower-than-nominal coverage (Bengio & Grandvalet, 2004). This miscoverage is especially pronounced when folds are not independent, as data points contribute to both training and evaluation. More recently, Bates et al. (2023) showed that even in modern settings, such intervals can severely misrepresent model uncertainty. They observed that stronger regularization mitigates this effect by providing CV estimator with fresher data across folds. In our work, we capitalize on this known limitation by using leave-fold-out replications to directly assess model consistency. This not only exposes the weaknesses of standard cross-validation but also highlights the robustness gains achieved through our proposed data regularization strategies.

**Curriculum Learning**  Curriculum Learning (CL) is a training paradigm inspired by the human learning process, where models are exposed to increasingly difficult examples (Bengio et al., 2009). It has shown benefits for convergence and generalization across domains. A recent causal analysis (Li et al., 2024) highlights that CL is more effective when early tasks reinforce decision patterns that remain valid throughout training. While originally studied in reinforcement learning, the underlying principle—aligning task difficulty with model capacity—can be extended to other learning settings. For instance, in computer vision, curriculum-based augmentation strategies that gradually increase corruption severity during training have gained attention (Lu & Lam, 2023; Choi & Kim, 2024).

## 3 METHODS

### 3.1 STOCHASTIC DATA REGULARIZATION

We explore data regularization through stochastic transformations applied during training, structured along three core dimensions: *selection policies* (e.g., noise injection, random augmentation pipelines), *combination policies* (e.g., jointly applying multiple data regularization strategies), and *scheduling policy* (e.g., curriculum learning-based approach).

**Selection Policies** Selection policies define stochastic mechanisms for perturbing training inputs, thereby inducing regularization without altering the underlying task. In this work, we consider two complementary strategies under this paradigm: direct **Noise Injection** and **Random Transformations** drawn from augmentation sets.

*Noise Injection* applies input-space corruption by sampling corruption parameters dynamically at each step. We define a noise operator $\nu(\cdot)$ such that:

$$\tilde{x} = \nu(x; \theta_t), \quad \theta_t \sim \mathcal{P}(\theta_{\min}, \theta_{\max}), \tag{1}$$

where $\mathcal{P}$ is a generic sampling distribution (e.g., Uniform, Alpha-Stable (Yuan et al., 2025)), and $\theta_t$ modulates the corruption strength (e.g., standard deviation for Gaussian noise or the corruption factor for Salt-and-Pepper noise). This technique is theoretically equivalent to Tikhonov regularization (Bishop, 1995) and is applied exclusively during training.

*Random Transformations*, in turn, select stochastic augmentations from a candidate set $\mathcal{T} = \{\tau_1, \tau_2, \ldots, \tau_k\}$. At each training step, a random subset $\mathcal{T}^* \subseteq \mathcal{T}$ is sampled and applied to the input with randomized parameters:

$$\tilde{x} = \tau(x), \quad \tau \in \mathcal{T}^*, \quad \text{with parameters from predefined ranges.} \tag{2}$$

**Combination Policies** In practice, multiple selection policies may be combined—e.g., applying both $\tau$ and $\nu$ sequentially—to form a compound perturbation strategy. Such combination policies can unify coarse (e.g., geometric, shuffling) and fine-grained (e.g., noisy) transformations, enabling richer training signals and broader robust generalization capabilities.

**Scheduling Policy** We implement a Curriculum Learning-based approach by progressively training across stages of increasing regularization, each governed by Early Stopping, as detailed in Algorithm 1. Each stage $s \in \{1, ..., S\}$ introduces a transformation $\Phi_s(x)$ selected from a predefined scheduling sequence $\mathcal{S} = (\Phi_1, ..., \Phi_S)$, which may include selection policies (e.g. Eq. 2, Eq. 1) or their combinations.

---

**Algorithm 1** Scheduling Policy with Early Stopping

---

**Input:** Training set $\{X^{train}, Y^{train}\}$, Validation set $\{X^{val}, Y^{val}\}$
**Input:** Scheduling sequence $(\Phi_1, ..., \Phi_S)$, Early Stopping patience values $(p_1, \ldots, p_S)$
**Input:** Hypothesis Space $\mathcal{H}$, Optimizer $\mathcal{U}$, Loss $\mathcal{L}$
**Output:** Final hypothesis $h_S \in \mathcal{H}$
1:  $h_1 \leftarrow h \in \mathcal{H}, \epsilon \leftarrow \infty$
2: **for** $s = 1$ to $S$ **do**
3:     $p \leftarrow 0, \tilde{h} \leftarrow h_s$
4:     $X_s^{train} \leftarrow \Phi_s(X^{train})$
5:     **while** $p > p_S$ **do**
6:        $\tilde{h} \leftarrow \mathcal{U}(X_s^{train}, Y^{train}, \tilde{h})$
7:        $\hat{Y}^{val} \leftarrow \tilde{h}(X^{val})$
8:        $\tilde{\epsilon} \leftarrow \mathcal{L}(\hat{Y}^{val}, Y^{val})$              ▷ Validation set used without transformations
9:        **if** $\tilde{\epsilon} < \epsilon$ **then**
10:           $\epsilon \leftarrow \tilde{\epsilon}, p \leftarrow 0, h_s \leftarrow \tilde{h}$
11:        **else**
12:           $p \leftarrow p + 1$
13:        **end if**
14:     **end while**
15:     $h_{s+1} \leftarrow h_s, s \leftarrow s + 1$
16: **end for**

---

## 3.2 CHARACTERIZING OUT-OF-DISTRIBUTION

We characterize out-of-distribution (OOD) datasets by estimating their divergence from the in-distribution data in a shared latent representation space (Algorithm 2). This approach avoids direct comparisons in the input space, which may be sensitive to superficial or non-semantic differences. We train an autoencoder ($\mathbf{h} \equiv (\mathbf{h}_f, \mathbf{h}_g)$) on the in-distribution training set and use its encoder ($\mathbf{h}_f$) to extract latent representations for both the clean test set ($\mathbf{Z}_{\text{in}}$) and each OOD variant ($\mathbf{Z}_{\text{out}}$). To quantify the shift between these distributions, we employ Kullback–Leibler (KL) divergence, although other statistical distances (e.g., Wasserstein) are compatible with our framework.

---

**Algorithm 2** Characterizing Out-of-Distribution Data using Latent Representations

---

**Input:** In-distribution dataset $X^{\text{in}} = \{x_i\}_{i=1}^{n_{\text{in}}}$, $x_i \in \mathbb{R}^d$, with training and test splits: $X^{\text{train}}, X^{\text{test}}$
**Input:** Out-of-distribution dataset $X^{\text{out}} = \{x_j\}_{j=1}^{n_{\text{out}}}$, $x_j \in \mathbb{R}^d$
**Input:** Learning Algorithm $\mathcal{A} : \mathbb{R} \to (\mathbf{h}_f, \mathbf{h}_g)$ where $\mathbf{h}_f : \mathbb{R}^d \to \mathbb{R}^p$ (encoder) and $\mathbf{h}_g : \mathbb{R}^p \to \mathbb{R}^d$ (decoder)
**Output:** Out-of-Distribution metric $M = \text{KL}(\mathbf{Z}_{\text{in}} \parallel \mathbf{Z}_{\text{out}})$
1: $\mathbf{h} \leftarrow \mathcal{A}(X^{\text{train}})$
2: $\varepsilon \leftarrow 10^{-10}$                                  ▷ Define a small positive constant
3: $\mathbf{Z}_{\text{in}} \leftarrow \mathbf{h}_f(X^{\text{test}})$                    ▷ Encode in-distribution test dataset $\mathbf{Z}_{\text{in}} \in \mathbb{R}^{n_{\text{in}} \times p}$
4: $\mathbf{Z}_{\text{in}} \leftarrow \text{flatten}(\mathbf{Z}_{\text{in}}) + \varepsilon$       ▷ Encode in-distribution test dataset $\mathbf{Z}_{\text{in}} \in \mathbb{R}^{n_{\text{in}} \times p} \times 1$
5: $\mathbf{Z}_{\text{in}} \leftarrow \frac{\mathbf{Z}_{\text{in}}}{\mathbf{1}^T \mathbf{Z}_{\text{in}}}$
6: $\mathbf{Z}_{\text{out}} \leftarrow \mathbf{h}_f(X^{\text{out}})$                   ▷ Encode out-of-distribution dataset $\mathbf{Z}_{\text{out}} \in \mathbb{R}^{n_{\text{out}} \times p}$
7: $\mathbf{Z}_{\text{out}} \leftarrow \text{flatten}(\mathbf{Z}_{\text{out}}) + \varepsilon$    ▷ Encode in-distribution test dataset $\mathbf{Z}_{\text{out}} \in \mathbb{R}^{n_{\text{out}} \times p} \times 1$
8: $\mathbf{Z}_{\text{out}} \leftarrow \frac{\mathbf{Z}_{\text{out}}}{\mathbf{1}^T \mathbf{Z}_{\text{out}}}$
9: $M = \mathbf{Z}_{\text{in}}^T \cdot \log \mathbf{Z}_{\text{in}} - \mathbf{Z}_{\text{in}}^T \log \mathbf{Z}_{\text{out}}$          ▷ Compute KL Divergence

---

## 3.3 MISCOVERAGE STATISTICAL ANALYSIS

We formalize our miscoverage evaluation via leave-fold-out analysis, detailed in Algorithm 3.

---

**Algorithm 3** Leave-Folds-Out Miscoverage Analysis

---

**Input:** Set of $K$-Fold Estimates $\mathcal{F} = \{\mathbf{F}_i : \mathbf{F}_i \in \mathbb{R}^B\}_{i=1}^K$
**Input:** Number $L$ of folds to leave out
**Output:** Set of Tuples $\mathcal{R} = \{(\tilde{\mu}_{\mathbf{R}_j}, \sigma_{\mathbf{R}_j})\}_{j=1}^J$
1: $J \leftarrow K - L$
2: $\mu_{\mathbf{F}} \leftarrow \frac{1}{N} \mathbf{1}^T \mathbf{F}$                                  ▷ Compute mean
3: $\mathcal{R} \leftarrow \emptyset$
4: **for** $j = 1$ to $J$ **do**
5:     $\mathbf{R}_j \leftarrow \mathbf{F} - \{\mathbf{F}_j, \ldots, \mathbf{F}_{j+L-1}\}$                   ▷ $\mathbf{R}_j \in \mathbb{R}^{B \times J}$
6:     $\tilde{\mu}_{\mathbf{R}_j} = \mu_{\mathbf{R}_j} - \mu_{\mathbf{F}}$                                  ▷ Mean-centered
7:     $\sigma_{\mathbf{R}_j} \leftarrow \text{bootstrap}(\mathbf{R}_j)$
8:     $\mathcal{R} \leftarrow \mathcal{R} \cup (\tilde{\mu}_{\mathbf{R}_j}, \sigma_{\mathbf{R}_j})$
9: **end for**

---

# 4 EXPERIMENTAL SETUP

Our experimental setup is designed to evaluate the impact of data regularization strategies on in-distribution, out-of-sample and out-of-distribution scenarios, as seen in Figure 3. All results reported in the main paper refer to the CIFAR-10 (Krizhevsky et al., 2009) dataset and its corrupted variant, CIFAR-10-C (Hendrycks & Dietterich, 2019). The CIFAR-10-C benchmark includes 19 corruption types (e.g. JPEG compression, contrast, brightness) across 5 severity levels, resulting in a total of 95 out-of-distribution datasets. We use F1-score as the evaluation estimator $\theta$ throughout all in-distribution and out-of-distribution assessments.

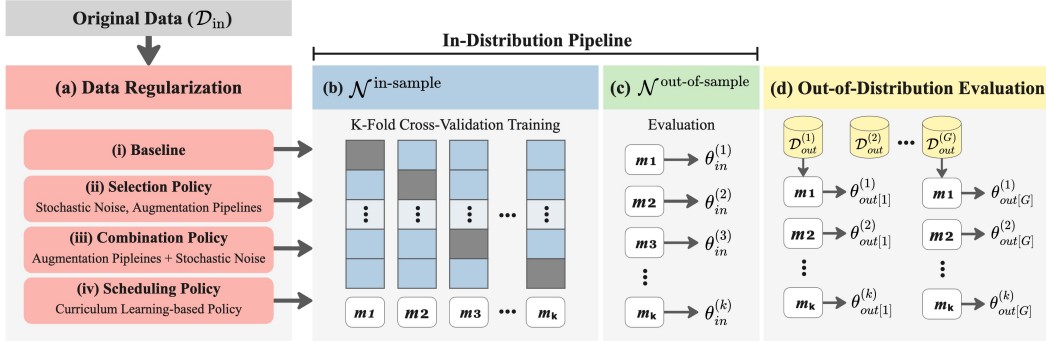

Figure 1: Overview of our evaluation pipeline. (**a**) We apply distinct stochastic data regularization strategies to the original in-sample dataset $\mathcal{D}_{in}$ for training, including no regularization (Baseline) and our modular framework consisting in 3 different policies—*selection*, *combination*, and *scheduling*—to regularize our data. (**b**) For each strategy, models are trained using $K$-fold cross-validation over $\mathcal{D}_{in}$. (**c**) Trained models are evaluated on $\mathcal{D}_{in}$ to obtain $\theta_{in}^{(i)}$ estimators. (**d**) Each model is then evaluated on a collection of $G$ corrupted datasets $\{\mathcal{D}_{out}^{(g)}\}_{g=1}^{G}$ to compute out-of-distribution estimators $\theta_{out[g]}^{(i)}$, enabling robust generalization and miscoverage analysis under domain shift.

## 4.1 ARCHITECTURES

We evaluate three representative architectures to assess the generalization effects of data regularization. **ResNet-20** (He et al., 2016) serves as a compact and shallow baseline with approximately 280K parameters. **WideResNet-28-10** (Zagoruyko & Komodakis, 2016) is a deeper and significantly wider CNN, totaling over 36M parameters, representing a high-capacity architecture. Finally, **CCT** (Compact Convolutional Transformer) (Hassani et al., 2021) introduces a hybrid transformer-based model with convolutional tokenization and positional encoding, comprising around 930K parameters. This setup allows us to contrast different architectural families—shallow CNNs, wide CNNs, and attention-based models—under a unified training protocol. All models are trained from scratch, without architectural-level regularization (e.g., Dropout or LayerNorm), to isolate the effects of data regularization alone. Inputs are 32×32×3 and training uses a batch size of 128.

## 4.2 IMPLEMENTATION DETAILS

Early Stopping was applied consistently across all training routines—including both standard and curriculum-based strategies—to prevent overfitting and stabilize convergence. Although this promotes fair evaluations, baseline models often converge prematurely, leading to lower final performance that may differ from typical state-of-the-art results. This is intentional, as our focus lies in understanding the robust generalization capabilities of models rather than maximizing absolute performance.

Standard training strategies used a fixed patience of 10 epochs, while Curriculum Learning stages followed a progressive patience schedule tailored to the difficulty of each stage. The same Early Stopping protocol was also applied to the Autoencoder used in the KL divergence characterization (see Section 3.2), ensuring consistent training dynamics across all components of the experimental pipeline.

## 4.3 DATA REGULARIZATION STRATEGIES

All stochastic data regularization strategies evaluated in this study are organized within our modular framework of **Selection Policies**, **Combination Policies**, and **Scheduling Policy**. Table 1 summarizes the configuration details, including the maximum Salt & Pepper factor, Gaussian noise standard deviation, RandAugment parameters, Curriculum Learning stage schedule, and Early Stopping values.

We apply **RandAugment** (Cubuk et al., 2020)—composed of both transformations (e.g. color jitter, Gaussian blur, and saturation adjustments) and standard augmentations like random cropping—as a representative *Selection Policy*, using three transformations per image with a fixed magnitude of 0.3. As a *Combination Policy*, we compose RandAugment with additive noise—either Gaussian or Salt & Pepper—to enhance perturbation diversity. At each training step, the noise intensity is dynamically sampled from a uniform distribution: $\sigma \sim \mathcal{U}(0, \sigma_{\max})$ for Gaussian noise, and $\lambda \sim \mathcal{U}(0, \lambda_{\max})$ for Salt & Pepper.

Finally, we implement a curriculum-based *Scheduling Policy*, where the regularization severity increases across training stages. All parameter values used in these strategies were selected through a lightweight parameter search. As a result, the Scheduling Policy focuses on the most effective configurations found—namely, RandAugment followed by RandAugment combined with Gaussian noise.

For a visual illustration of the applied perturbations under each strategy, see Appendix A. The GitHub repository will be made publicly available for full reproducibility.

Table 1: Configurations for CIFAR-10 training strategies, including data regularization types, noise levels, and Early Stopping (ES) patience.

| Policy | Strategy | Max S&P Factor ($\alpha$) | Max Gaussian StdDev ($\sigma$) | ES Patience |
|---|---|---|---|---|
| Baseline | None | – | – | 10 |
| Selection Policy | RandAugment | – | – | 10 |
| Combination Policy | RandAugment + S&P | 0.3 | – | 10 |
| Combination Policy | RandAugment + GN | – | 0.2 | 10 |
| Scheduling Policy | **Stage 1:** RandAugment | – | – | 3 |
| | **Stage 2:** RandAugment + GN | – | 0.1 | 5 |
| | **Stage 3:** RandAugment + GN | – | 0.2 | 8 |

## 5 RESULTS

To enable severity-aware comparisons, we characterize each of the 95 CIFAR-10-C corruptions by measuring their divergence from the clean CIFAR-10 distribution in latent space and calculating the KL Divergence. This is performed using Algorithm 2. Based on the resulting metric vector, we sort all corruptions and divide them into three severity bands according to their percentile rank: **Lowest** (0–33rd), **Mid-Range** (34–66th), and **Highest** (67–100th). Figure 2 shows this categorization, with average KL values and bootstrapped confidence intervals per group. This

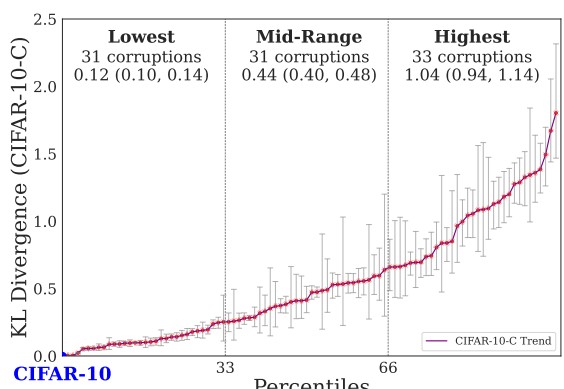

Figure 2: KL-based categorization of CIFAR-10-C corruptions.

severity stratification underpins all subsequent robustness analyses presented in this study.

To evaluate the effectiveness of data regularization strategies, we report F1-scores under both in-distribution (out-of-sample) and out-of-distribution (OOD) scenarios. As shown in Figure 5 and Table 2, augmenting training with stochastic regularization significantly improves performance across all models and severity levels. In particular, we observe that applying RandAugment improves OOD generalization compared to the baseline. Furthermore, combining RandAugment with noise yields even stronger improvements under high-severity corruptions. Finally, the Scheduling Policy consistently achieve superior F1-scores under stronger corruptions, especially for lightweight architectures like ResNet20.

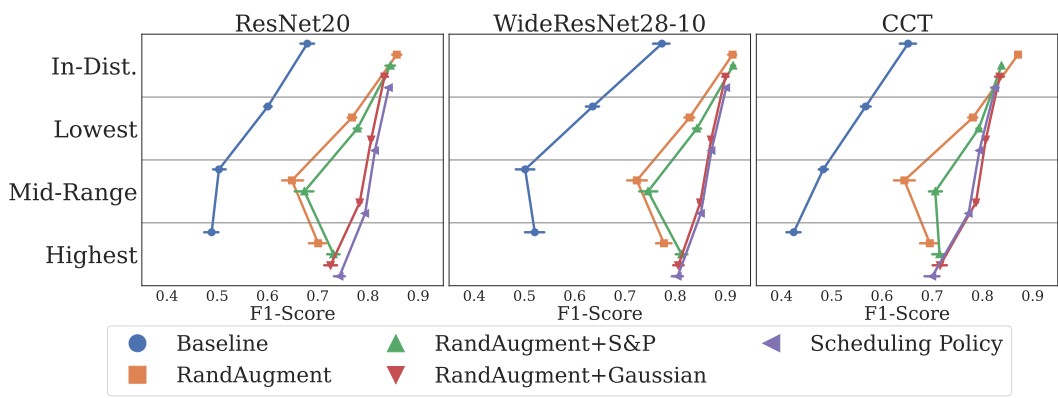

Figure 3: F1-scores for each model under In-Distribution and Out-of-Distribution (OOD) scenarios, with corruptions grouped by severity (Lowest, Mid-Range, Highest).

Table 2: Out-of-Distribution (OOD) characterization for CIFAR-10-C grouped by severity. Values are F1-score (95% CI). Epochs are average values. Best results per column are in bold.

| ResNet20 | In-Dist. | Lowest | Mid-Range | Highest | Epochs (avg) |
|---|---|---|---|---|---|
| Baseline | 67.9 (66.6, 69.4) | 60.1 (59.3, 60.9) | 50.4 (49.1, 51.6) | 48.9 (47.5, 50.3) | 15.4 |
| RandAugment | **85.7 (84.8, 86.7)** | 76.8 (75.9, 77.8) | 64.9 (62.8, 66.8) | 70.1 (68.5, 71.8) | 51.4 |
| RandAugment+S&P | 84.5 (83.6, 85.5) | 77.9 (77.2, 78.7) | 67.4 (65.6, 69.2) | 73.2 (72.0, 74.4) | 65.3 |
| RandAugment+Gaussian | 83.3 (82.6, 84.0) | 80.6 (80.3, 81.0) | 78.4 (77.9, 78.8) | 72.6 (71.4, 73.9) | 63.0 |
| Scheduling Policy | 84.2 (83.6, 84.8) | **81.4 (81.1, 81.8)** | **79.4 (79.1, 79.8)** | **74.3 (73.3, 75.5)** | 91.6 |
| **WideResNet-28-10** | | | | | |
| Baseline | 77.4 (75.7, 79.2) | 63.6 (62.4, 64.9) | 50.1 (48.2, 52.0) | 52.1 (50.3, 53.8) | 19.0 |
| RandAugment | 91.4 (90.7, 92.3) | 82.9 (81.9, 83.8) | 72.4 (70.5, 74.4) | 77.8 (76.3, 79.3) | 53.6 |
| RandAugment+S&P | **91.5 (91.1, 92.0)** | 84.3 (83.5, 85.1) | 74.6 (72.9, 76.4) | **81.3 (80.1, 82.6)** | 56.6 |
| RandAugment+Gaussian | 90.0 (89.4, 90.7) | **87.2 (86.8, 87.6)** | 85.0 (84.6, 85.5) | 80.6 (79.6, 81.7) | 59.9 |
| Scheduling Policy | 90.2 (89.7, 90.7) | **87.2 (86.8, 87.6)** | **85.1 (84.7, 85.5)** | 80.5 (79.4, 81.8) | 62.1 |
| **CCT** | | | | | |
| Baseline | 65.2 (63.6, 66.8) | 56.8 (55.8, 57.8) | 48.3 (47.3, 49.4) | 42.4 (41.1, 43.8) | 15.6 |
| RandAugment | **87.1 (86.7, 87.5)** | 78.1 (77.2, 79.1) | 64.5 (62.4, 66.7) | 69.5 (67.7, 71.5) | 95.0 |
| RandAugment+S&P | 83.8 (83.6, 84.0) | 79.3 (78.8, 79.8) | 70.6 (69.4, 71.8) | 71.5 (70.1, 73.0) | 100.0 |
| RandAugment+Gaussian | 83.5 (82.6, 84.4) | **80.7 (80.3, 81.1)** | **78.7 (78.3, 79.1)** | **71.6 (70.3, 73.1)** | 99.1 |
| Scheduling Policy | 82.5 (81.8, 83.3) | 79.4 (79.0, 79.9) | 77.3 (76.8, 77.7) | 70.0 (68.5, 71.6) | 77.7 |

Table 3: Standard deviation (95% CI) of F1-scores across CIFAR-10-C severity ranges. Lower values indicate more stable performance across folds. Best results per column are in bold.

| ResNet20 | In-Dist. | Lowest | Mid-Range | Highest |
|---|---|---|---|---|
| Baseline | .0237 (.0229, .0244) | .0718 (.0707, .0728) | .1144 (.1134, .1154) | .1337 (.1321, .1354) |
| RandAugment | .0139 (.0135, .0143) | .0643 (.0636, .0650) | .1418 (.1409, .1427) | .1292 (.1278, .1307) |
| RandAugment+S&P | .0157 (.0149, .0164) | .0670 (.0660, .0681) | .1603 (.1590, .1616) | .1162 (.1140, .1184) |
| RandAugment+Gaussian | .0106 (.0101, .0112) | .0313 (.0305, .0319) | .0389 (.0385, .0394) | .1173 (.1140, .1206) |
| Scheduling Policy | **.0098 (.0094, .0103)** | **.0306 (.0299, .0313)** | **.0338 (.0334, .0341)** | **.1048 (.1016, .1081)** |
| **WideResNet-28-10** | | | | |
| Baseline | .0273 (.0255, .0293) | .1167 (.1150, .1186) | .1658 (.1646, .1671) | .1695 (.1679, .1710) |
| RandAugment | .0106 (.0103, .0110) | .0674 (.0668, .0681) | .1394 (.1386, .1403) | .1192 (.1180, .1206) |
| RandAugment+S&P | **.0072 (.0068, .0076)** | .0736 (.0725, .0747) | .1635 (.1622, .1647) | .1128 (.1108, .1148) |
| RandAugment+Gaussian | .0111 (.0107, .0116) | .0356 (.0348, .0365) | .0427 (.0422, .0432) | **.1002 (.0979, .1030)** |
| Scheduling Policy | .0075 (.0068, .0081) | **.0343 (.0335, .0353)** | **.0399 (.0395, .0404)** | .1039 (.1006, .1068) |
| **CCT** | | | | |
| Baseline | .0250 (.0237, .0261) | .0881 (.0866, .0895) | .0897 (.0889, .0906) | .1317 (.1303, .1330) |
| RandAugment | .0094 (.0090, .0099) | .0588 (.0581, .0595) | .1316 (.1306, .1325) | .1490 (.1472, .1507) |
| RandAugment+S&P | **.0041 (.0039, .0043)** | .0439 (.0432, .0447) | .1136 (.1124, .1147) | **.1319 (.1287, .1350)** |
| RandAugment+Gaussian | .0143 (.0136, .0150) | **.0349 (.0342, .0357)** | **.0363 (.0359, .0367)** | .1332 (.1299, .1365) |
| Scheduling Policy | .0113 (.0107, .0119) | .0377 (.0369, .0385) | .0400 (.0395, .0405) | .1340 (.1307, .1373) |

To complement average performance results, we apply our proposed Miscoverage Analysis (Algorithm 3) to assess the stability of model predictions across cross-validation folds. Figure 5 visualizes miscoverage behaviors, while Table 3 reports the standard deviation and 95% confidence intervals of F1-scores across severity categories, highlighting the consistency gains from data regularization strategies.

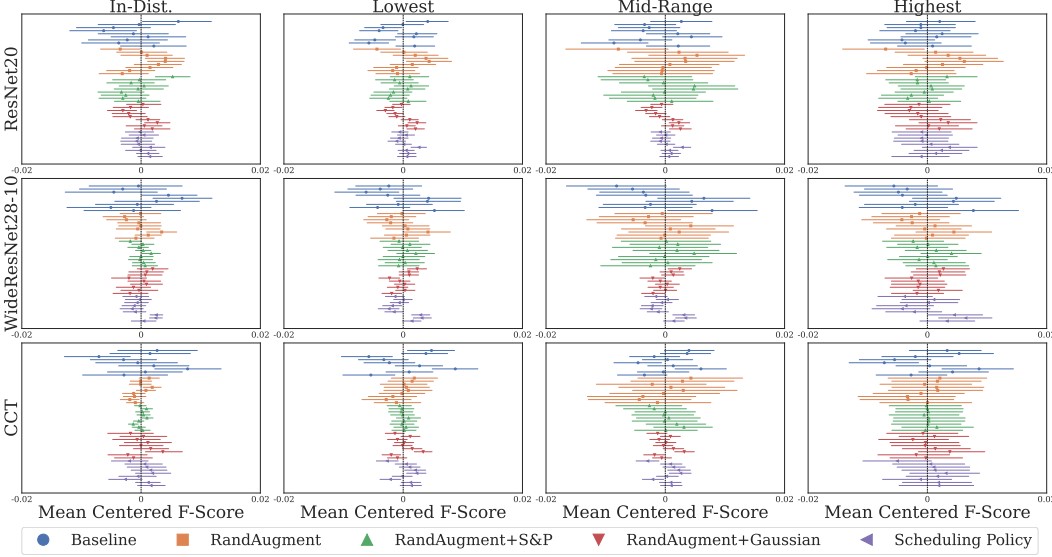

Figure 4: Mean-Centered F-Score distributions for each data regularization strategy across three architectures (ResNet20, WideResNet-28-10, and CCT) and four evaluation domains: In-Distribution and OOD corruptions grouped by severity levels. Each point represents a mean-centered F1-score from one replication, and horizontal lines denote 95% confidence intervals derived via bootstrap resampling. These results are generated using the Leave-Folds-Out Miscoverage Analysis (Algorithm 3). Narrower intervals suggest more stable generalization across unseen corruptions and folds.

## 6 Discussion

Our findings underscore that robust generalization is strongly influenced by how data regularization is designed and scheduled during training. Rather than relying solely on isolated techniques, **our modular framework offers a structured lens for understanding and improving robust generalization**, especially under out-of-distribution shifts.

Scheduling Policies (Curriculum Learning-based) consistently with state-of-the-art proves effective in enhancing robustness across architectures and corruption severities. It reliably reduces performance variability (i.e., lower standard deviations) and often outperforms isolated regularization strategies. These findings emphasize the importance of the training data presentation order for generalization under distribution shift.

Combining stochastic data regularization techniques also yields measurable benefits. In particular, pairing RandAugment with Gaussian noise improves robustness for WideResNet and CCT, whereas Salt & Pepper noise produces less consistent gains. This suggests that the interaction between model architecture and corruption type is nontrivial and deserves further attention.

Notably, improvements are not solely driven by the type of regularization applied, but also by increased training exposure. Curriculum-based strategies tend to train longer before Early Stopping is triggered. This extended exposure appears necessary for learning robust representations, indicating that vision models benefit from prolonged stochastic regularization.

Some of the applied data regularization strategies may partially overlap with corruptions present in the CIFAR-10-C dataset. To understand the effects under this lens, we include a discussion in Appendix B comparing overall performance on the full CIFAR-10-C benchmark and a filtered version that excludes overlapping corruptions—specifically, cases where similar transformations (e.g., Gaussian noise or contrast adjustments introduced by RandAugment) are used during training but are also present in the evaluated corrupted test sets.

In summary, our results highlight that the structure and progression of regularization—not just its presence—play a critical role in enabling robust generalization, particularly for compact models. A tailored and modular approach to data regularization, as proposed in this study, offers a promising direction for building more reliable machine learning systems under distributional stress.

## 7 Conclusion

This work explored stochastic data regularization strategies to improve model robustness under distribution shift. We find that organizing these transformations into a curriculum—progressively increasing complexity during training—consistently leads to more stable and generalizable models. While curriculum-based approaches demonstrate strong regularization capabilities, especially under challenging conditions, we also observe that simpler strategies combining transformations with noise injection offer competitive trade-offs in terms of effectiveness and efficiency. These results highlight that the structure and dynamics of data exposure can be as important as the regularization technique itself. For future work, we aim to explore a Variational Autoencoder to extract a structured latent space, thereby enhancing our quantification of distribution shifts. Additionally, we intend to apply the methods proposed in this work to other data modalities, such as acoustic signals and text.

### Acknowledgments

Not applicable.

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

## A  DATA REGULARIZATION TRANSFORMATIONS

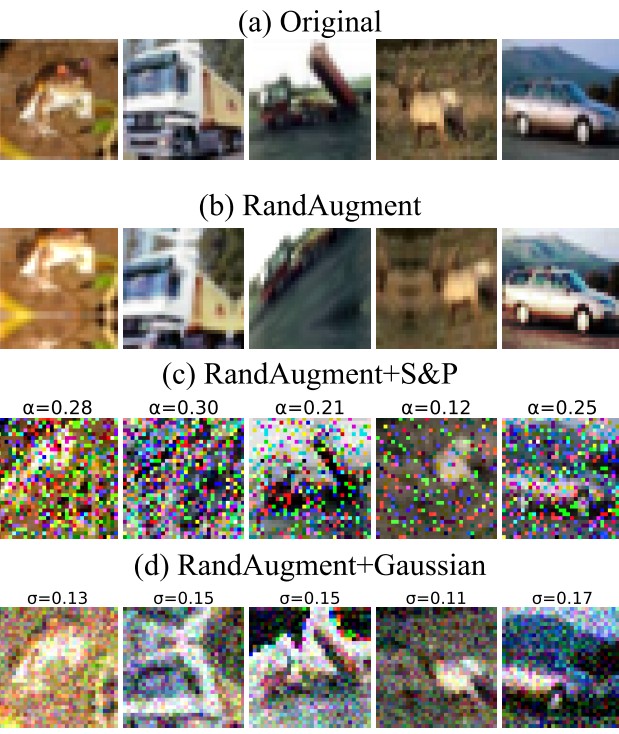

Figure 5: Visual comparison of stochastic data regularization strategies applied to CIFAR-10 samples. (**a**) Original images. (**b**) RandAugment with 3 transformations per image and magnitude 0.3. (**c**) RandAugment combined with Salt & Pepper noise (*Combination Policy*), where the noise factor $\alpha$ is sampled uniformly from $\mathcal{U}(0, \alpha_{\max} = 0.3)$. (**d**) RandAugment combined with Gaussian noise, with standard deviation $\sigma \sim \mathcal{U}(0, \sigma_{\max} = 0.2)$. The values of $\alpha$ and $\sigma$ shown below each image indicate the sampled noise intensity for that example.

## B  OVERALL RESULTS ACROSS ALL CORRUPTIONS

Some of the data regularization strategies we adopt—such as Gaussian noise, Salt & Pepper noise, and RandAugment—introduce transformations that can be thought of as partially overlapping with

specific corruptions in the CIFAR-10-C dataset (e.g., *Gaussian Noise*, *Shot Noise*, *Speckle Noise*, *Impulse Noise*, *Contrast*, *Brightness*). We conducted a sensitivity analysis comparing a dataset with all curruptions included and a dataset (*w/o Overlap*) that excludes these potentially overlapping curruptions. The (*w/o Overlap*) dataset is, when compared to the dataset containing all corruptions, OOD to a greater extent. The results remain consistent, and the exclusion of potentially overlapping corruptions has even improved in some cases.

Table 4: General performance for three models using different augmentation strategies. Values are F1-score (95% CI). Best results per column are in bold.

| ResNet20 | In-Dist. | All Corruptions | w/o Overlap |
|---|---|---|---|
| Baseline | 67.9 (66.5, 69.4) | 53.0 (52.3, 53.9) | 55.2 (54.5, 56.0) |
| RandAugment | **85.7 (84.8, 86.7)** | 70.6 (69.7, 71.5) | 73.3 (72.3, 74.2) |
| RandAugment+S&P | 84.5 (83.6, 85.5) | 72.8 (72.0, 73.6) | 75.0 (74.4, 75.7) |
| RandAugment+Gaussian | 83.3 (82.6, 84.0) | 77.1 (76.6, 77.6) | 76.8 (76.4, 77.2) |
| Scheduling Policy | 84.2 (83.6, 84.8) | **78.3 (77.9, 78.8)** | **78.1 (77.7, 78.5)** |
| **WideResNet-28-10** | | | |
| Baseline | 77.4 (75.7, 79.4) | 55.2 (54.2, 56.3) | 57.3 (56.2, 58.4) |
| RandAugment | **91.4 (90.7, 92.3)** | 77.7 (76.8, 78.6) | 79.6 (78.6, 80.6) |
| RandAugment+S&P | **91.5 (91.1, 92.0)** | 80.1 (79.3, 80.9) | 81.1 (80.2, 82.0) |
| RandAugment+Gaussian | 90.0 (89.3, 90.7) | **84.2 (83.7, 84.7)** | **83.5 (83.1, 84.0)** |
| Scheduling Policy | 90.2 (89.8, 90.7) | **84.2 (83.7, 84.7)** | **83.6 (83.2, 84.1)** |
| **CCT** | | | |
| Baseline | 65.2 (63.7, 66.8) | 49.0 (48.3, 49.8) | 50.1 (49.3, 50.9) |
| RandAugment | **87.1 (86.6, 87.5)** | 70.7 (69.6, 71.7) | 75.3 (74.5, 76.2) |
| RandAugment+S&P | 83.8 (83.5, 84.0) | 73.7 (73.0, 74.5) | 75.4 (74.9, 76.0) |
| RandAugment+Gaussian | 83.5 (82.6, 84.4) | **76.9 (76.3, 77.5)** | **76.8 (76.4, 77.2)** |
| Scheduling Policy | 82.5 (81.9, 83.2) | 75.4 (74.8, 76.1) | 75.3 (74.8, 75.8) |

## C ACKNOWLEDGMENT OF LLM USE

We acknowledge the use of large language models to aid in polishing the writing and, primarily, to help build and check how tables and plots could be presented in the best way. The models were not used to generate original research content, experiments, or results.

