# OpenReview forum: "Towards Robust Out-of-Distribution Generalization for Deep Neural Networks with Tailored Data Regularization"
_ICLR.cc/2026/Conference — Submitted to ICLR 2026_

### Official Review · Reviewer_oidv · 2025-10-15

**Soundness:** 2
**Presentation:** 3
**Contribution:** 2
**Rating:** 2
**Confidence:** 3

**Summary:**

This paper systematizes probabilistic data regularization methods aimed at improving DNN robustness to perturbations and empirically evaluates a curriculum-inspired scheduling approach. It further proposes a new severity metric for corruptions based on the KL divergence between internal representations of an autoencoder, and introduces an evaluation protocol intended to overcome the miscoverage issues inherent to standard K-fold CV. Experiments on CIFAR-10 and CIFAR-10-C with three model families show that stronger regularization yields better generalization under the proposed evaluation, and that the proposed scheduling approach outperforms other baselines only for the relatively small model (ResNet20).

**Strengths:**

- Systematization of probabilistic data regularization: The paper organizes existing common regularizers into three application modes—selective application, combination, and curriculum-style scheduling of regularization strength—an organization that seems practically useful.
- KL-based difficulty measure for corruptions: The paper quantifies perturbation distance from the training distribution using KL divergence computed over an autoencoder’s internal representations.
- Toward a more essential evaluation protocol: The proposed protocol attempts to address coverage reliability rather than focusing solely on average performance.

**Weaknesses:**

- Limited novelty: Robustness via data regularization has a long history, and its effectiveness is widely known. While a systematic combination and evaluation of heterogeneous regularizers is less common, the paper does not demonstrate performance gains large enough to make this viewpoint a substantial contribution on its own.
- Narrow experimental scope: Validation is restricted to CIFAR-10/CIFAR-10-C. Demonstrating sizable improvements on broader datasets (e.g., CIFAR-100, ImageNet) would make the claims more solid.
- Unconvincing results for scheduling: The curriculum-style regularization shows benefits primarily for ResNet20. On WideResNet-28-10 and CCT, the scheduling approach is not clearly superior—and sometimes underperforms—non-scheduling variants. As a result, the main takeaway appears to be simply that “using regularization helps,” rather than that curriculum-style scheduling is consistently better.

**Questions:**

- Performance appears roughly correlated with training epochs; the success of scheduling on ResNet20 might be due to longer training rather than the curriculum mechanism in itself. What happens under a fixed-epoch comparison?

- Beyond F1, incorporating calibration metrics (e.g., ECE, Adaptive ECE) would make the evaluation more compelling.

- Some figure/table references seem incorrect:
  - Line 211: Figure 3 → Figure 1

  - Line 318: Figure 5 → Figure 3

  - Line 401: Figure 5 → Figure 4

---

### Official Review · Reviewer_yQiq · 2025-10-29

**Soundness:** 2
**Presentation:** 2
**Contribution:** 1
**Rating:** 2
**Confidence:** 5

**Summary:**

This paper proposes a data augmentation framework for OOD generalization. The proposed framework comprises three ingredients: selection policies that stochastically select data transformations, combination policies that combine selected data transformations sequentially, and a scheduling policy that serves as a curriculum learning framework. The proposed method is evaluated on CIFAR-10-C.

**Strengths:**

n/a

**Weaknesses:**

There is little technical contribution in this paper. In the actual implementation, "selection policy" uses RandAugment as is, "combination policy" combines RandAugment and Gaussian/Salt-and-Pepper noise injection, and "scheduling policy" is a standard curriculum learning setup based on pre-defined early stopping criteria. Moreover, adding the final scheduling policy results in little performance gains on CIFAR-10-C, which is itself a small-scale OOD generalization benchmark that should not be used alone for benchmarking. I think the paper at its current stage is far from the bar of ICLR.

**Questions:**

see Weaknesses

---

### Official Review · Reviewer_NLBW · 2025-10-31

**Soundness:** 1
**Presentation:** 2
**Contribution:** 1
**Rating:** 2
**Confidence:** 5

**Summary:**

The paper proposes a tailored data augmentation strength strategy to improve OOD classification of deep networks in OOD cases. The paper evaluates two convolutional networks on CIFAR-10C.

**Strengths:**

**S1.** The paper is tackling a relevant problem: improving OOD classification performance.

**S2.** The paper is well-written and easy to follow. **

**Weaknesses:**

**W1.** Lack of novelty: The methodology presented here strongly resembles the curriculum learning strategy from [1], which is not even cited in the paper despite its significant similarity (see the next weakness).

[1] EfficientNetV2: Smaller Models and Faster Training. ICML 2021

**W2.** Missing related work: The related work section is extremely poor, with as few as 8 references, considering the importance of topics such as out-of-distribution categorization/curriculum learning, and its wide popularity within the field.

**W3.** Missing previously published methodology: The analysis of section 3.2 (Algorithm 2) to characterize distance among ID and OOD samples does not employ previously published and well-known approaches for this task [2].

[2] OoD-Bench: Quantifying and Understanding Two Dimensions of Out-of-Distribution Generalization. CVPR 2021

**W4.** Narrow experimental setup: The empirical tests are performed on a single dataset (CIFAR10) and corresponding corrupted versions (CIFAR10C), not meeting expectations from an empirical perspective for a conference paper at ICLR.

**W5.** Figures' layout (pages 7 and 8) is not suitable for a conference of the ICLR level.

**Questions:**

I appreciate the authors’ effort and recognize the relevance of the topic.

However, the main concerns (limited novelty, missing literature coverage, and lack of usage of previously published methodology) reflect substantive rather than clarificatory issues. Therefore, I do not have specific questions for rebuttal that would likely alter my assessment.
In this form, I see the paper as more suitable for a workshop or a minor conference rather than ICLR main conference.

---

### Official Review · Reviewer_H32E · 2025-11-01

**Soundness:** 2
**Presentation:** 3
**Contribution:** 2
**Rating:** 4
**Confidence:** 3

**Summary:**

This paper addresses the challenge of out-of-distribution (OOD) generalization in deep neural networks through a modular framework for stochastic data regularization. The framework combines selection, combination, and scheduling policies to dynamically adapt data perturbations during training. By organizing noise injection and random transformations under a curriculum learning scheme, the method gradually increases perturbation strength to align model capacity with task difficulty. The paper also introduces a miscoverage-based statistical analysis to quantify the stability and reliability of performance across cross-validation folds. Experiments show that the proposed regularization scheduling improves OOD robustness and generalization stability.

**Strengths:**

1. The paper is well written and organized.
2. The proposed framework is comprehensive.
3. The author has taken a wide range of experiments to verify this method.

**Weaknesses:**

1. The frameworks seems as a combination of some known ideas, such as RandAugment, noise injection, and curriculum learning, which shows limited novelty.
2. Then experiments mainly focus on CIFAR-10, could you verify this idea on more diverse datasets and tasks?
3. Can the proposed framework integrate or compare against adversarial training or spectral regularization techniques?
4. How sensitive is the scheduling policy to hyperparameters like the stage count or early stopping patience?

**Questions:**

See weakness.

---

### Meta-Review · Area_Chair_q1ZY · 2026-01-07

**Summary:**

The paper presents a framework for improving Out-of-Distribution (OOD) generalization in Deep Neural Networks through a modular data regularization approach. The method integrates selection, combination, and scheduling policies (utilizing a Curriculum Learning scheme) to dynamically adjust noise injection and transformations during training. Additionally, it proposes a KL-divergence-based metric for assessing corruption severity and a statistical procedure to evaluate performance consistency across cross-validation folds.

**Reviewer Concerns:**

The reviewers primarily highlighted the lack of technical novelty in combining existing techniques (RandAugment, curriculum learning), the narrow experimental scope restricted to CIFAR-10, and inconsistent results across larger model architectures.

**Reviewer Scores:**

The consensus was negative, with one score of 4 and three scores of 2, reflecting a collective assessment that the paper is not yet ready for the main conference.

---

### Decision · Program_Chairs · 2026-01-26

Reject